# Towards Health Status Determination and Local Weather Forecasts from *Vitis vinifera* Electrome

**DOI:** 10.3390/biomimetics10090636

**Published:** 2025-09-22

**Authors:** Alessandro Chiolerio, Federico Taranto, Giuseppe Piero Brandino

**Affiliations:** 1Bioinspired Soft Robotics, Istituto Italiano di Tecnologia, Via Morego 30, 16065 Genova, Italy; 2Unconventional Computing Laboratory, University of the West of England, Coldharbour Lane, Bristol BS16 1QY, UK; 3Hopeful Futures, 1507EL Zaandam, The Netherlands; 4eXact-Lab srl, Via Crispi 56, 34126 Triste, Italy; brandino@exact-lab.it

**Keywords:** *Vitis vinifera*, bioelectric potential, electrophysiology, machine learning, weather forecast, bioinspiration, *Flavescence dorée*

## Abstract

Recent advances in plant electrophysiology and machine learning suggest that bioelectric signals in plants may encode environmentally relevant information beyond physiological processes. In this study, we present a novel framework to analyse waveforms from real-time bioelectrical potentials recorded in vascular plants. Using a multi-channel electrophysiological monitoring system, we acquired continuous data from *Vitis vinifera* samples in a vineyard plantation under natural conditions. Plants were in different health conditions: healthy; under the infection of *Flavescence dorée*; plants in recovery from the same disease; and dead stumps. These signals were used as input features for an ensemble of complex machine learning models, including recurrent neural networks, trained to infer short-term meteorological parameters such as temperature and humidity. The models demonstrated predictive capabilities, with accuracy comparable to sensor-based benchmarks between one and two degree Celsius for temperature, particularly in forecasting rapid weather transitions. Feature importance analysis revealed plant-specific electrophysiological patterns that correlated with ambient conditions, suggesting the existence of biological pre-processing mechanisms sensitive to microclimatic fluctuations. This bioinspired approach opens new directions for developing plant-integrated environmental intelligence systems, offering passive and biologically rooted strategies for ultra-local forecasting—especially valuable in remote, sensor-sparse, or climate-sensitive regions. Our findings contribute to the emerging field of plant-based sensing and biomimetic environmental monitoring, expanding the role of flora to biosensors, useful in Earth system observation tasks.

## 1. Introduction

The increasing demand for high-resolution, real-time weather forecasting has catalyzed the exploration of unconventional data sources and methodologies. Traditional meteorological models, while robust, often lack the granularity required for hyper-local predictions, especially in heterogeneous terrains and microclimates. In turn, weather forecasting can be profitably used to predict crop yield by exploiting similar ML tools [1]. The link between meteorology and biology extends to the scale of the predictive models, and represents a profound connection between two apparently detached fields [2]. Recent advancements in plant electrophysiology suggest that plants, through their bioelectrical signals, may serve as sensitive indicators of environmental changes, offering a novel avenue for localized weather forecasting [3]. Plants exhibit a range of electrical activities in response to environmental stimuli, including variations in temperature, humidity, light intensity, and mechanical stress [4]. These bioelectrical responses [5], encompassing action potentials and variation potentials, are integral to plant signalling mechanisms and have been documented to reflect external environmental conditions [6]. Recent studies have approached the causal chain in a statistical way, highlighting that most of plants respond with specific bioelectric signals in a consistent and repeatable way to stimuli given [7]. Electrome dynamics exhibit non-random behaviour, temporal correlations, and persistence [8], suggesting a potential role in long-distance electrical signalling in individual plants under conditions like osmotic stimuli and temperature variations. The potential use of these signals as proxies for environmental monitoring has been further underscored by studies demonstrating their correlation with specific abiotic stressors, as discussed in [9] and in a recent work [10], where further references are included. Electrophysiological parameters collected from plants have even been used to track specific infections, such as *Plasmodiophora* in the case of *Brassica rapa* [11]. Ongoing research is currently developing means to realize less invasive interfaces with plant tissues [12].

The integration of machine learning (ML) techniques with plant electrophysiological data has opened new frontiers in environmental sensing. ML algorithms, particularly deep learning models, have shown proficiency in deciphering complex non-linear patterns within bioelectrical signals, enabling the classification and prediction of various environmental parameters [13]. For instance, supervised ML models have been employed to predict vines’ water status based on electrophysiological inputs, achieving notable accuracy [14]. In another study, machine learning algorithms (Artificial Neural Networks, Convolutional Neural Network, Optimum-Path Forest, k-Nearest Neighbors and Support Vector Machine) have been combined with Interval Arithmetic, and the findings show that Interval Arithmetic and supervised classifiers are more suitable than deep learning techniques [15].

Building upon this foundation, the present study explores the feasibility of utilizing plant bioelectrical signals as inputs for ML models aimed at forecasting local weather conditions. By harnessing the innate sensitivity of plants to their immediate environment, we propose a biomimetic approach to weather prediction that complements existing meteorological methods. This interdisciplinary endeavor aligns with the principles of biomimetics, wherein biological systems inspire innovative technological solutions.

The objectives of this research are threefold: (1) to establish a reliable methodology for recording and processing plant bioelectrical signals in situ; (2) to develop and train ML models capable of translating these signals into accurate short-term weather forecasts; and (3) to evaluate the performance of these models against conventional forecasting techniques. With this study, we aim to contribute to the development of sustainable, plant-based environmental monitoring systems that enhance the precision of local weather forecasting.

## 2. Materials and Methods

### 2.1. Experimental Setup and Crop Features

A traditional vineyard plant was selected to perform this study, as shown in Figure 1. It is located in Vigliano d’Asti, Monferrato, Italy (Cantina Adorno). For the collection of bioelectric potentials, two needle electrodes (MN4022D10S subdermal electrodes from SPES MEDICA SRL, Genova, Italy) 0.4 mm in diameter and 22 mm in length were used to contact each plant by puncturing the bark. They were positioned 1 cm apart and connected using a double-shielded ultra-low-resistance cable INCA1050HPLC (MD, Atri (TE) Italy) for high-fidelity audio application to a DI-710-US data logger (DATAQ Instruments, Akron, OH, USA), used in differential channel mode, recording at a frequency of 1 sample/s per channel, with 16 bit resolution and 1000 mV of voltage range. The maximum cable length was 5.2 m, adding an additional resistance component of 36 ± 2 mΩm^−1^, an additional capacitance component of 78 ± 2 pF mm^−1^, and an additional inductive component of 78 ± 400 nH m^−1^. The cable structure is composed by an inner copper core, a first polyethylene dielectric insulation, a second semiconductive insulation, a third metallic net shield with coverage of 90%, and a final polyvinylchloride flexible coverage. The presence of the semiconductive insulation between the dielectric one and the net shield allows us to discharge accumulated charges due to triboelectrification using cable movements, generally responsible for additional noise. The ideal electrical connection scheme is shown in the inset of Figure 1. Local climatic data was provided by a probe station (Primo Principio S.r.l., Alghero (SS), Italy). The eight plants object of this study were selected as statistically representative of the entire plantation—a high-density vineyard. Furthermore, a logistic aspect was considered in the choice of the sample, with close plants allowing for the use of a single recording device connected by the shortest possible cables to avoid signal corruption by long transmission lines. The input from dead, affected, and healthy plants is considered to be equally important. The plants’ health status was determined on the basis of manned observation by agronomers. Here, the discriminant is the action induced on the plants by *Flavescence dorée*, which has become an endemic disease in the Monferrato area where the vineyard is located; therefore, their symptoms, as well as the aspect of a plant that survives to the phytoplasm and recovers until fruiting again, are very well known. The causal agent of *Flavescence dorée* is a phytoplasm belonging to the group of grapevine yellows, which colonizes the host’s phloem tissues and causes blockage of the elaborated sap, triggering an imbalance in the plant’s physiological activities. In late-stage manifestations, the grape clusters progressively shrivel and may partially or completely dry out; in cases of early symptom onset, infected shoots appear gummy in texture and tend to bend downwards, giving the plant a prostrate appearance [16]. It first appeared in 2000. Unfortunately, there is no known remedy for the *Flavescence dorée*; the current action taken by farmers and enforced by local governments consists of spreading insecticides that should terminate *Scaphoideus titanus*, known as being responsible for the spreading of the disease. Recovery occurs as a combination of agronomic actions (pruning infected branches) and, eventually, of plant reinforcement through other branches. This process typically requires 3–4 years before a fruiting season appears, unless the plant dies in between. After infection, the plants develop a very peculiar structure: even after recovery, the signs of *Flavescence dorée* can be recognized, as their bark becomes more permeable and fragile, while healthy plants preserve a neat structure. Therefore, by ’healthy plants’ we mean individuals which have never been infected, and by ’plants under recovery’ we mean individuals infected between 1 and 4 years before, too sick to bear fruits at all, and by ’fully recovered plants that have produced fruits’ we mean individuals infected between 20 and 4 years before. Collection site ‘A’ represents a plant under recovery phase with no fruits yet, ‘B’ a plant after recovery bringing fruits, ‘C’ a dead stump, ‘D’ a plant after recovery bringing fruits, ‘E’ a plant manifesting the symptoms of *Flavescence dorée*, ‘F’ a dead stump, ‘G’ a plant after recovery bringing fruits, and ’H’ a healthy individual. Interestingly, the qualitative features of biopotential signals well-characterize the health status of the vines, and we can easily cluster the “cleaner” waveforms collected from healthy plants and plants undergoing full recovery that bring fruits apart from dead stumps still showing an elaborated electrical activity (vines maintain living roots for years even when the photosynthetic part has gone) and apart from vines that show the typical symptoms of *Flavescence dorée*. Lastly, we have positioned a cut log of a vine on a plastic insulating foil above ground, in the same location, using the same setup described above, to collect additional control data, out of something that is definitely dead and cannot host any spontaneous bioelectric activity (identified as site “I”). Statistical analysis has been performed using OriginPro 2022 software.

### 2.2. Electrome Correlation Analyses and Spectral Features

The acquired data is shown in Figure 2, after being smoothed using a first-order Savitzky–Golay function with a window size of 49 points. The different individuals feature always a bioelectric granularity different than the control sample (black curve “I”; colour code kept throughout the paper). Statistical data is presented on the following figures, starting with the histogram analysis of Figure 3. The broadest voltage distribution belongs to the sample featuring the symptoms of *Flavescence dorée*, a symmetrical Gaussian-like set (“E”). The healthy individual features a narrow, asymmetric distribution (“H”). The control sample features the most narrow distribution (“I”). A boxplot is shown in Figure 4, where the voltage span measured from plants is seen in comparison to the control measure in black.

After discussions about individual signal analyses, we aim to show potential correlations between electrome signals, highlighting the correlations between different plants is fundamental to understand how redundant is data fed into the ML model. Reducing the input allows us to invest less time and raw materials. Furthermore, in plant health monitoring, correlated electrical responses can precede the visible symptoms of stress (drought, pathogens, nutrient deficiency) [17]. In dense plant communities (greenhouses, forests, crops), correlated electrome patterns might indicate collective behaviours—shared defence responses, synchronized flowering signals, or ecosystem-level stress adaptation [18]. Strong correlations mean more reliable signal processing, enabling plants themselves to become part of a cyber–physical sensing infrastructure [19].

Figure 5 shows a comparison between two correlation frameworks: the Pearson’s and the Spearman’s. The first correlation measures how much two independent processes are linear, while the second one measures how much two independent processes are monotonic, and works better in situations where nonlinearities are found, as well as in situations where strong outliers are present. Green dashed curves have been added to highlight the rows where the different individuals sit, giving readers an easy indication of possible clusterings between neighbouring plants (for example, individuals F, G, and H in the far row, which show a positive correlation, and individuals D and E in the middle row, which show a negative correlation). *p*-values analysis allows us to exclude only two correlations, which are not significant, between plant A and plants E and H. Spearman’s correlations show a highly similar pattern, with some differences in numerical values; the most remarkable aspect is that the *p*-value analysis allows us to exclude another couple, plant A and F, and confirms as valid the previous two reported above. We can witness, overall, a good degree of interdependency between a specific plant couple: F and G with a positive correlation of 0.94513 (0.81865) according to Pearson (Spearman). Less impacting correlations are found in the same row between G and H, with a positive coefficient of 0.35986 (0.33909), and between F and H, with a positive coefficient of 0.35989 (0.31792). The intermediate row shows a correlation between E and D, with a negative coefficient of −0.3823 (−0.31994). The first row shows correlations, whose absolute value is below 0.3. The only significant correlation outside the rows is found between B and D, with a negative coefficient of −0.3728 (−0.33021). The similarity between Pearson and Spearman’s correlations, with slightly higher values in the first case, suggests more linearity among the measured processes. In light of this analysis, we suggest that individuals F and G might have developed a strong physical anastomosys, with probable infiltration of the radical system of the dead stump by means of the living plant.

Visualizing the signals in the frequency domain provides us with other relevant information on the different behaviours of the plants, as shown in Figure 6. In that figure, we plot the power spectral density of each signal as Time-Integral Squared Amplitude (TISA) versus frequency of the Fast Fourier Transform (FFT), in bilogarithmic scale. We note that the plant under recovery “A” has a unique flat distribution of the fluctuations over the entire spectrum of measured frequencies; the plants that have undergone recovery and are fruiting again, “B”, “D”, and “G”, have in common a trait where the TISA is inversely proportional to the frequency, and a flat trait at higher frequencies, much like the healthy individual “H”, whose cutoff frequency is, however, smaller. The plant with ongoing symptoms of *Flavescence dorée* “E” has two distinctive features: the highest TISA at higher frequencies, and preserves a descending trait where the TISA is inversely proportional to signal frequency, at the highest frequency range we have measured. The two dead stumps, “C” and “F”, also feature very high values of TISA, as well as a less evident descending trait at high frequencies. Finally, the cut log of a vine features the lowest TISA in almost every region of the spectrum.

### 2.3. Attempting to Resolve the Health Status by Means of Signal Features

By combining all of the information presented above, and carefully considering the existing correlations between signals, in particular between vines F and G, it is possible to attempt resolving the health condition of each plant. Stump F’s features are identical to vine G in recovery status with fruits, with extremely high positive meaningful correlations (Figure 5), and we can consider this stump to be governed in its electrical response from vine G itself. Besides this notable exception, we found a relevant set of parameters to identify the health statuses of the individuals tested, namely: statistical distribution skewness s (if s < −1 “<<”; if −1 < s < −0.1 “<”; if s > −0.1 “o”), Kurtosis k (if k > 1 “>”; if k < 1 “o”), histogram range within 1.5 IQR r (if r > 20 mV “+++”, if 20 < r < 10 mV “+”, if r < 10 mV “−”), median (if Med > 0 “+”, if Med < 0 “−”), the cardinal relation between average and median (as per Figure 4, if the mean square is above the median line “>”, if it is below “<”, if it is across the line “=”), and FFT features (as per visual inspection of Figure 6). Results are summarized in Table 1.

Below is a key to cluster the biopotentials and identify the health status unambiguously:A healthy plant is characterized by the unique features of having a small range of values, a negative median value, and a highly negative skewness; the small range could be due to tight homeostatic control of water status and ion fluxes that keeps excursions modest. The negative median baseline is slightly “below zero” vs. reference and could be due to tension-dominated xylem potential, mildly hyperpolarized tissues, or electrode offset); the highly negative skewness could be explained by the fact that many small fluctuations with occasional sharper downward events occur (nighttime recharge, brief stomatal closures, cavitation micro-events [20]), producing a long negative tail, while positive bursts are rare.A plant which has undergone recovery is characterized by the unique features of having a small range, a negative median value, a slightly negative skewness, and a flat FFT. The small range and negative median indicate that homeostasis is mostly restored, the baseline is still on the negative side, the skewness is only slightly negative (the large negative tail has subsided), there are fewer acute stress dips, and there are reduced spectral peaks (flatter spectrum) indicating damped oscillations post-stress [21].A plant which has undergone recovery and brings fruits is characterized by the unique features of having a positive median value; the fruit load shifts source–sink relations, sustained phloem loading, and turgor dominate the measured signal (e.g., more depolarized tissues/greater pressure), pushing the baseline above zero; sugar transport and anabolic activity bias the distribution to the positive side, even if variability stays modest [22].A plant which features ongoing *Flavescence dorée* is characterized by the unique features of having near-zero skewness and Kurtosis, a very broad range, and a peculiar three-stage FFT. The broad range might be due to phytoplasma-driven phloem dysfunction (callose deposition), hormonal imbalance, and carbohydrate backlog, which causes alternating episodes of hydraulic/electrical instability; large excursions in both directions; near-zero skewness and Kurtosis; variability is symmetric and mesokurtic, erratic but not dominated by single-sided bursts or heavy tails; and a “Three-stage” FFT, where distinct bands emerge from (i) slow phenological/metabolic drift, (ii) diurnal transpiration–stomatal cycles decoupled by pathology, and (iii) fast stress/compensatory oscillations from disrupted sieve-tube transport and ion flux feedbacks [23].A dead stump is characterized by the unique features of having an extended range and a negative median value. The extended range can be explained by the fact that, without living regulation, the sensor tracks have unbuffered environmental swings (temperature, humidity, soil moisture, EM noise), so spread explodes; the negative median is due to persistent desiccation/ionic drift or an electrode half-cell offset, which keeps the baseline on the negative side in the absence of an active metabolism [10].A control vine log is characterized by the unique features of having a negative skewness, a zero Kurtosis, a small range, and a negative median value. The small range and mesokurtic tails are typical of stable, near-Gaussian fluctuations around a steady setpoint; the negative skewness and negative median are like regulated baselines, as in healthy vines.

We observe that, if the vine under characterization is only one, or if the mutual correlations among individuals in a plantation are not analysed, wrong conclusions might be reached.

### 2.4. Software Setup

The data collected was sorted in different comma-separated values files. The saved data was loaded in Python 3.9, preprocessed by filtering the quantile 0.8 of the frequency domain of the signal. The 8 signals were aggregated from 1 s to 1 h granularity using their median values. Eventually, the 8 signals were merged with the temperature data for model (1) and with humidity data for model (2), both at 1 h granularity. Then, the min–max scaler was fitted to transform each file separately. The ML algorithm selected for the data analysis is the Long Short-Term Memory (LSTM) Recurrent Neural Network [24], with peephole augmentation [25], wrapped by a Bidirectional layer. The LSTM requires 3 dimensions as the input shape; with number of samples, number of observations (number of sequences), and number of features (number of voltage readings), we rearranged the data with a window of 48 h, so as to be able to identify the circadian cycles of the temperature for each batch of data. We used the same approach to predict relative humidity in the air. Therefore, with N datapoints of voltage readings, we obtained a train shape of (N-48, 48, 8), while the validation size has been the floor of 0.2 times N-48. The files that did not have at least 48 datapoints as the validation dataset in the first dimension were discarded. The final shapes are (321, 48, 8) for the validation data and (321, 48) for the validation target, while they are (984, 48, 8) for the training data and (984, 48) for the training target. The ML problems defined as (1) and (2) attempt to predict the (1) temperature and (2) humidity for the current hour (t) given 48 h of previous data from time t-1 to time t-48. Through different trials, we identified the following architecture for the Tensorflow layers:
Model: "sequential"
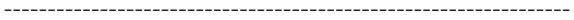
 Layer (type)Output ShapeParam #
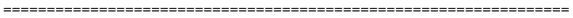
 bidirectional (Bidirectional)(None, 48, 32)3296 multi_head_attention (MultiHeadAttention)(None, 48, 32)4224 dense (Dense)(None, 48, 32)1056 dropout (Dropout)(None, 48, 32)0 dense_1 (Dense)(None, 48, 1)33
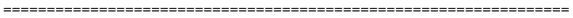
Total params: 8609 (33.63 KB)Trainable params: 8609 (33.63 KB)

The problem was implemented as a regression, and the loss used was the mean squared error. The training was performed by minimizing the validation loss (the loss of the validation dataset), and also the $R^{2}$ coefficient was collected. The learning rate policy was started at 0.00005 and increased to reach 0.01 in the last epoch. The number of epochs was set to a maximum of 1200, and an early stopping criterion was defined, which restored the model’s weights to those of the best performing iteration occurred. For a more detailed description, see the CombinedLRScheduler class in the codebase (https://github.com/bormiopoli/bio-ml/tree/main (accessed on 21 September 2025)).

## 3. Results

Both model (1) and (2) were assessed with validation data. The loss (Mean Squared Error) of the model minimized is relative to normalized values ranging from 1 to 0. The Pearson and Spearman correlations are 0.72 and 0.66 for humidity and 0.78 and 0.7 for temperature. The Pearson and Spearman *p*-values are, respectively, 7.33 × 10^−59^ and 7.08 × 10^−53^ for the temperature, while 3.22 × 10^−41^ and 1.69 × 10^−33^ for the humidity. The Mean Absolute Error is computed eventually on the validation dataset with non-standardised values to provide the reader with a more immediate understanding of the magnitude of error. The predictions shown herein were produced with time-continuous validation data originating from three of the eleven comma-separated files used (the remaining files did not have sufficient contiguous observations to append to the validation dataset). The first chunk is relative to data from 8 June 2023 to 9 June 2023. In Figure 7, model (1) predictions are plotted in blue against the observed temperature data in orange.

In Figure 8, model (2) predictions are plotted in blue against the observed air humidity data in orange.

The second chunk is taken from 8 July 2023 to 9 July 2023. Model (1) produced the results depicted in Figure 9, featuring the following prediction (blue) for the observed temperature data (orange).

For the same dates, model (2) produced the results depicted in Figure 10, featuring the following prediction (blue) for the observed relative air humidity data (orange).

The third chunk is taken from 14 July 2023 to 15 July 2023. Model (1) produced the results depicted in Figure 11, featuring the following prediction (blue) for the observed temperature data (orange).

For the same dates, model (2) produced the results depicted in Figure 12, featuring the following prediction (blue) for the observed relative air humidity data (orange).

The mean squared error (loss) relative to the standardised data is plotted along each iteration on the training dataset. Figure 13 and Figure 14 represent the loss for temperature and humidity standardised data, respectively.

The mean has been computed for the mean absolute errors (MAEs) computed on the same non-standardised data of the validation dataset (originating from all the 11 files used). The average MAE obtained on the validation data is around 1.66 °C for temperature and 7.35% for humidity. As a comparison, the typical accuracy for sensor-based (data assimilation) short-term weather forecasting and nowcasting is typically around 1–2 °C for temperature [26] and around 5–10% for humidity [27].

## 4. Conclusions and Future Prospects

This study demonstrates the feasibility of using bioelectrical signals from *Vitis vinifera* as a rich, biologically grounded source of environmental information. By employing a robust data acquisition system and applying advanced machine learning models, we successfully predicted short-term meteorological parameters, such as temperature and relative humidity, with promising accuracy. The electrophysiological signals were shown to be influenced both by the plant’s health status and by the physical correlations between different individuals and contained time-sensitive patterns aligned with rapid weather transitions, suggesting an innate environmental encoding within plant biopotentials.

Our findings support the concept that plants act as living sensors, exhibiting sensitivity to microclimatic fluctuations that can be computationally decoded. The high predictive performance of recurrent and attention-based models highlights the potential of biologically rooted weather forecasting systems, especially valuable in locations lacking dense sensor networks or in climate-sensitive agricultural regions.

Future work will focus on scaling up this study by incorporating a wider biodiversity of plant species and environmental conditions to generalize the models. Testing a larger population would eventually confirm, and ultimately allow us to establish, the electrophysiological features that vines possess in their different health statuses. Integration of multi-modal data (e.g., optical, thermal, chemical) alongside electrophysiology could enhance predictive granularity. Lastly, we envision the development of low-power embedded hardware systems for real-time processing and edge inference, enabling the deployment of biohybrid weather stations. These advancements could revolutionize ultra-local environmental monitoring by fusing biological intelligence with modern AI techniques.

## Figures and Tables

**Figure 1 biomimetics-10-00636-f001:**
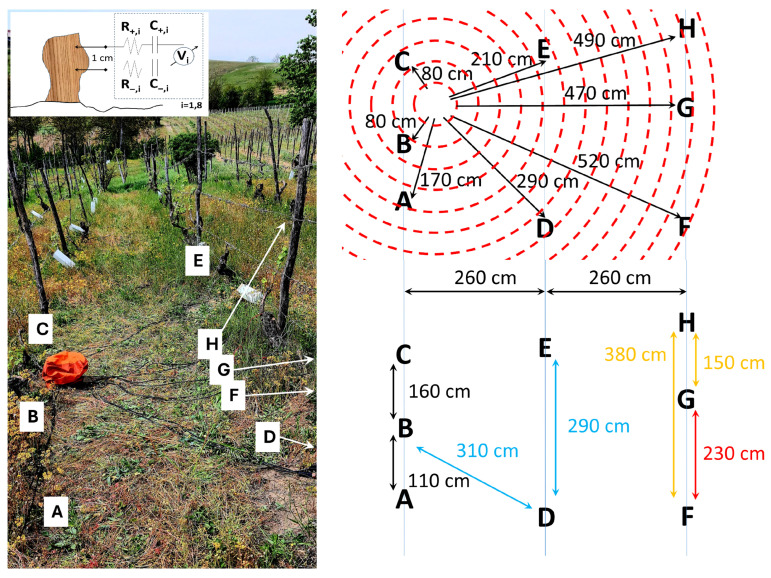
(**Left**) Measurement setup showing the traditional vineyard where *Vitis vinifera* plants were connected to the biopotential acquisition device (battery-powered, protected by the orange waterproof cover). Double-shielded high-fidelity cables are also shown running across the row. The setup was located in Cantina Adorno, Vigliano d’Asti (Monferrato, Italy). In the top inset, the ideal electrical connection scheme is shown, where two out of eight positive and negative leads and their related probes, series resistances ($R_{+/-,i}$) and series capacitances ($C_{+/-,i}$), and measurement channel ($V_{i}$) are shown. (**Top right**) Connection scheme where plants connected are identified using capital letters, rows are indicated by the light blue lines, the data logger measurement device is at the centre of the red disks, cables are identified by black arrows with their length reported in cm. (**Bottom right**) Distance and correlation scheme where plants are identified using capital letters, their distance and the inter-row distance in cm are evidenced by arrows, and a colour code is adopted on the basis of the correlation results (black: correlation ρ<25%; orange: positive correlation 25%<ρ<40%; light blue: negative correlation 25%<ρ<40%; red: positive correlation ρ>90%.

**Figure 2 biomimetics-10-00636-f002:**
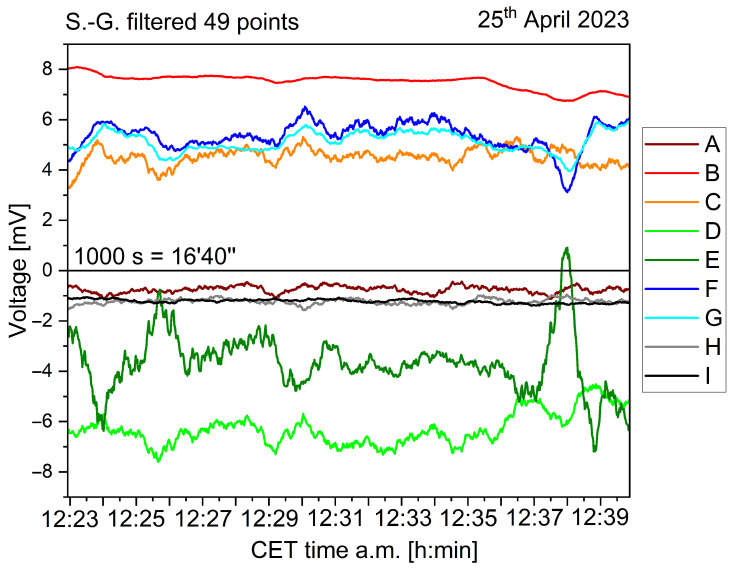
Biopotential recordings taken from eight different individuals of *Vitis vinifera* (coloured curves from A to H) and a control chunk (black curve I), over a time lapse of 1000 s (16 min 40 s) randomly selected within the entire dataset.

**Figure 3 biomimetics-10-00636-f003:**
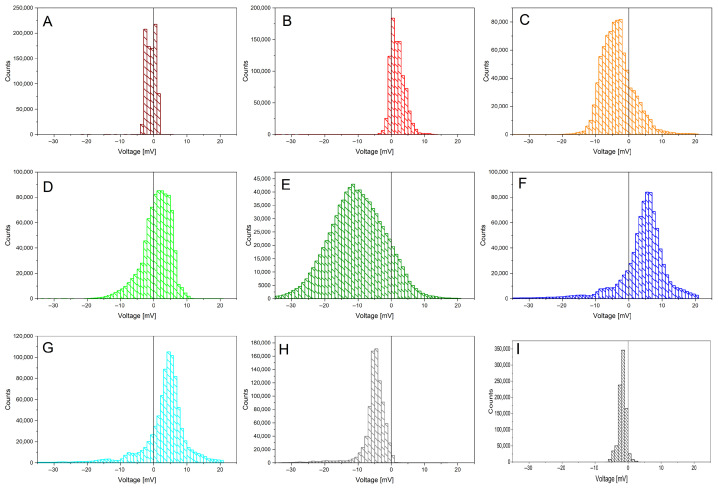
Statistical features of *Vitis vinifera* biopotentials recordings (coloured curves from (**A**–**H**)) and a control chunk (black curve (**I**)), over a time lapse of 242 h. Histograms showing distributions with respect to a zero potential (all values expressed in mV).

**Figure 4 biomimetics-10-00636-f004:**
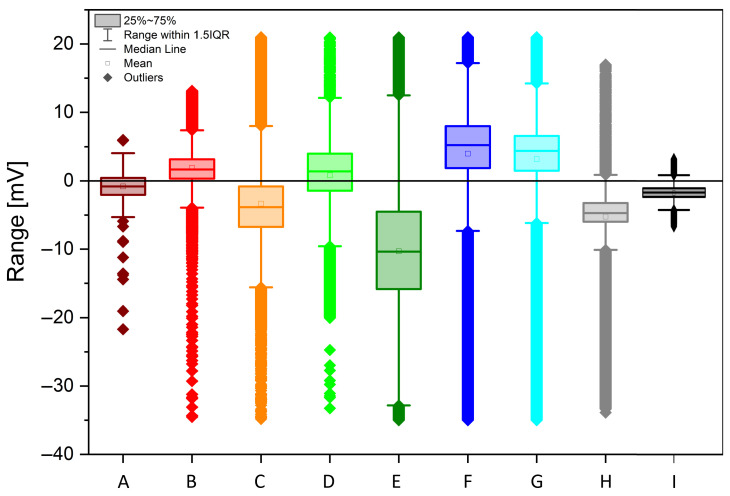
Statistical features of *Vitis vinifera* biopotentials recordings (coloured curves from A to H) and a control chunk (black curve I), over a time lapse of 242 h. Boxplot showing, for each process, the outliers (full rhomboids), interquartile population (filled rectangles), extended interquartile range corresponding to the 150 % span of the previous quantity (limited segment), median (horizontal segment), and mean (open square).

**Figure 5 biomimetics-10-00636-f005:**
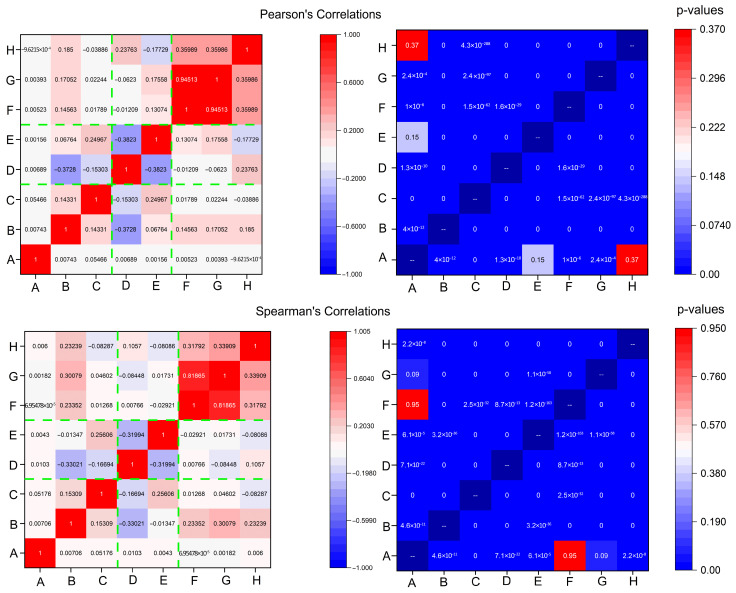
Statistical features of *Vitis vinifera* biopotentials recordings over a time lapse of 242 h. From top to bottom, from left to right: heatmap showing Pearson’s correlation coefficients (the green dashed lines group plants located in the same vineyard row), related *p*-values, Spearman’s correlations coefficients (the green dashed lines group plants located in the same vineyard row), and related *p*-values.

**Figure 6 biomimetics-10-00636-f006:**
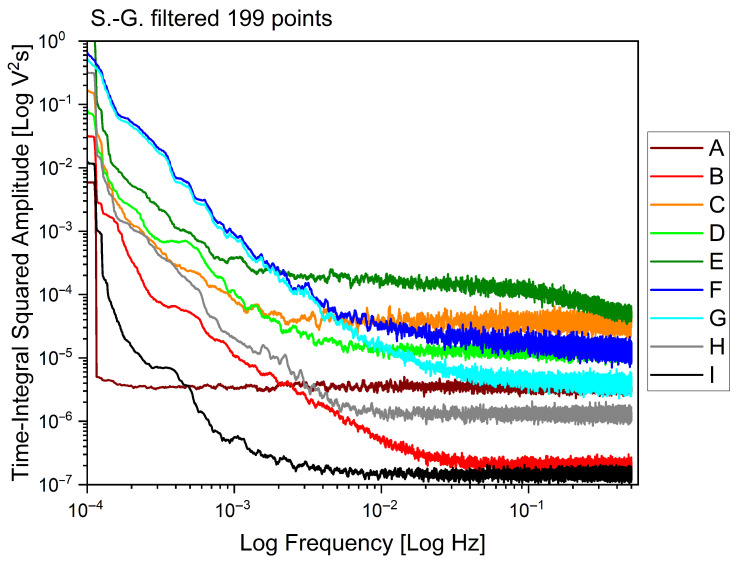
Fast Fourier Transform (FFT) analysis of the dataset, comparing the different sites from “A” to “H” and the cut log of a vine “I”. Data shows the logarithm of the Time-Integral Squared Amplitude (TISA) versus the logarithm of the frequency, filtered applying a Savitzki–Golay first-order function over 199 experimental points.

**Figure 7 biomimetics-10-00636-f007:**
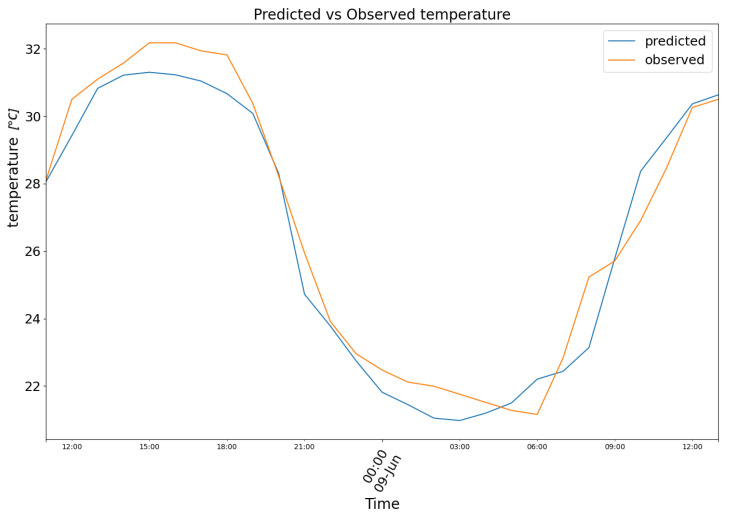
Predicted (blue) vs. Observed (orange) temperature.

**Figure 8 biomimetics-10-00636-f008:**
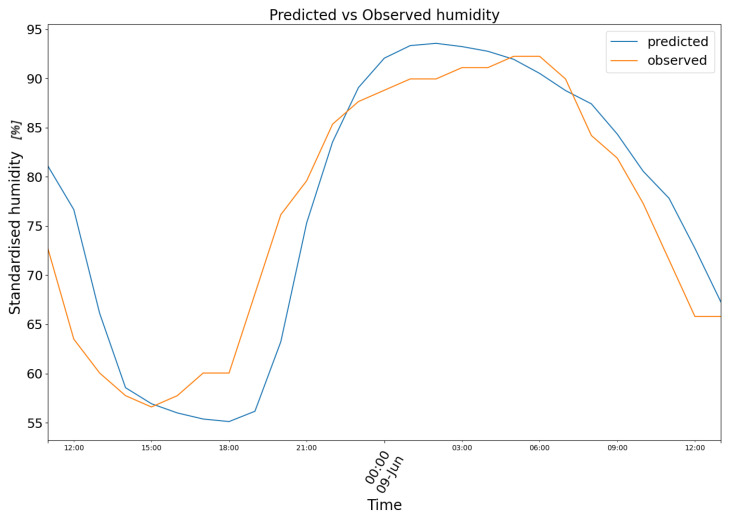
Predicted (blue) vs. Observed (orange) humidity.

**Figure 9 biomimetics-10-00636-f009:**
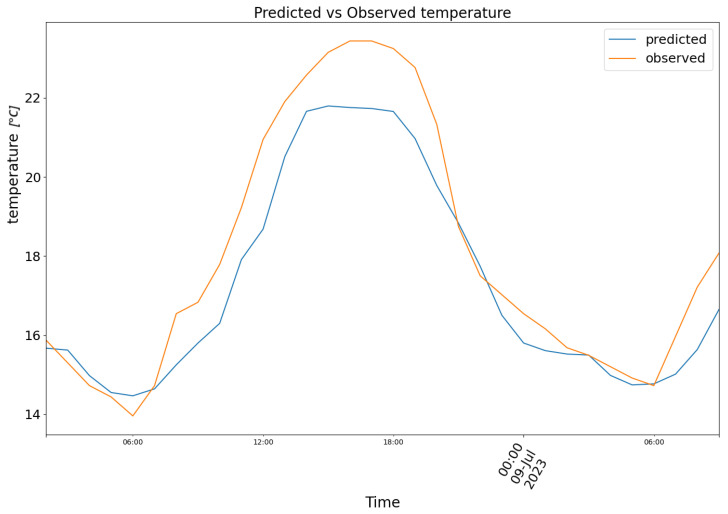
Predicted (blue) vs. Observed (orange) temperature.

**Figure 10 biomimetics-10-00636-f010:**
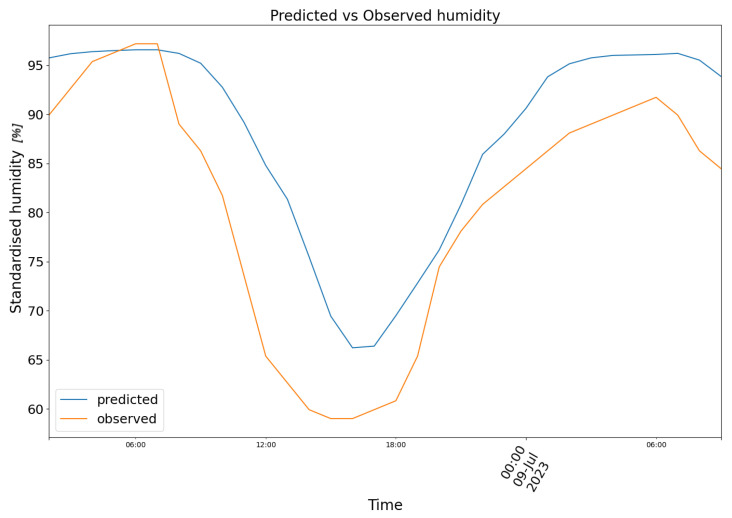
Predicted (blue) vs. Observed (orange) humidity.

**Figure 11 biomimetics-10-00636-f011:**
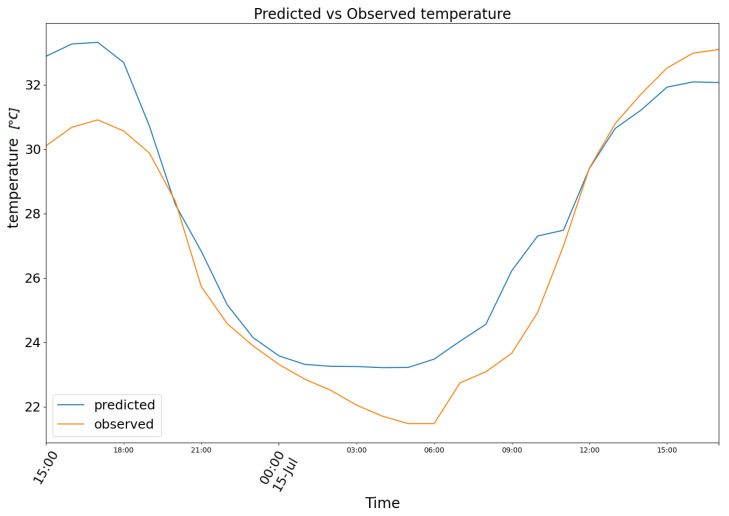
Predicted (blue) vs. Observed (orange) temperature.

**Figure 12 biomimetics-10-00636-f012:**
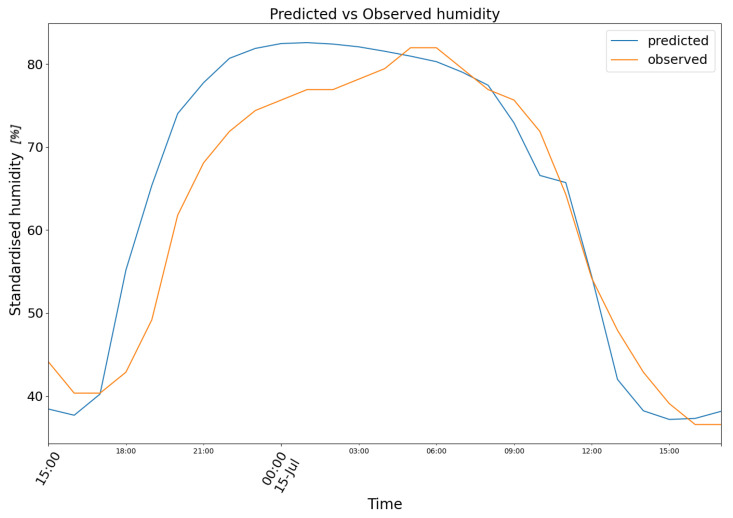
Predicted (blue) vs. Observed (orange) humidity.

**Figure 13 biomimetics-10-00636-f013:**
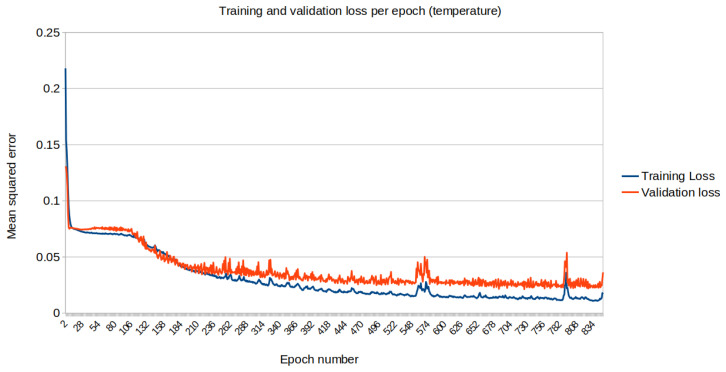
Training (blue) vs. Validation (orange) dataset loss for temperature.

**Figure 14 biomimetics-10-00636-f014:**
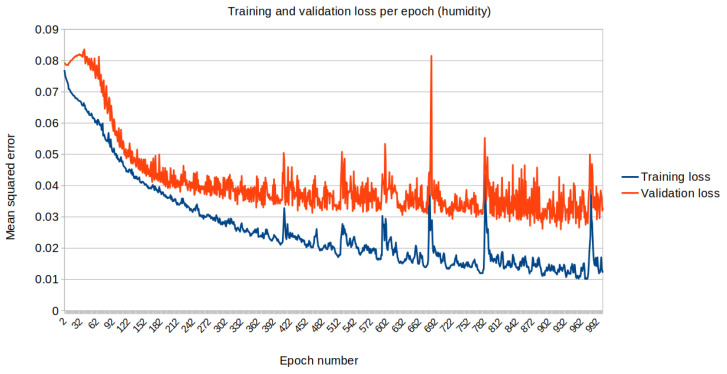
Training (blue) vs. Validation (orange) dataset loss for air humidity.

**Table 1 biomimetics-10-00636-t001:** Electrome parameters extracted from plants in different health statuses.

ID	Health Status	Skewness	Kurtosis	Range	Med	Avg/Med	FFT
A	recovery	<	>	−	−	=	flat
B	recovery + fruits	>	>	−	+	>	lin + flat
C	stump	>	>	+	−	>	lin + flat
D	recovery + fruits	<	>	+	+	<	lin + flat
E	flavescent	o	o	+++	−	=	lin + flat + lin
F	stump	<<	>	+	+	<	lin + flat
G	recovery + fruits	<<	>	+	+	<	lin + flat
H	healthy	<<	>	−	−	<	lin + flat
I	control	<	o	−	−	=	lin + flat

## Data Availability

Data will be made publicly accessible via Dryad.

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
