# Peer review of "Towards Health Status Determination and Local Weather Forecasts from Vitis vinifera Electrome"

_biomimetics, 2025, doi:10.3390/biomimetics10090636_

Round 1
Reviewer 1 Report (Previous Reviewer 1)
Comments and Suggestions for Authors
This is a resubmitted manuscript “Towards Local Weather Forecasts from Vitis vinifera Electrome” by Alessandro Chiolerio et al. I reviewed the initial submission and recommended it to be rejected by Biomimetics. In my opinion, the resubmitted work is vastly improved, the majority of my comments have been addressed. Despite all the positives, I still have a few observations.
General comments:
Methods
I still fail to find the authors’ interpretation of the meaning behind the specific correlations distinguished. Is it important that the signals from the last row of vines (F, G, H) are clustered, but it is not the case for the first row (A, B, C)? What was the scientific question that the authors solved using these correlations?
The authors added another figure (Fig. 5) to justify the claim that “the qualitative features of biopotential signals well characterize the health status of the vines, and we can easily cluster the "cleaner" waveforms collected from healthy and plants undergone full recovery, that bring fruits, from dead stumps still showing an elaborated electrical activity“ (page 3).
Looking at the data (In Fig. 5 and others), I can agree that the control trace (I) and the diseased plant trace (E) qualitatively differ from other traces. However, I still see no convincing evidence that it is possible to reliably differentiate between the plants of different health status, especially between dead stumps and recovering plants. In their response to my previous comments, the authors claimed that “The clustering might be influenced by the roots in anastomosys”, which may be true, but then one cannot be sure which trace belongs to the dead stump, and which trace belongs to another plant (for example, vines G and F). The frequency analysis, while commendable, also shows that, for example, the signal amplitude from the healthy plant (H) is higher than the signal amplitude from B (recovering plant) for all frequencies. At the same time, the signal amplitude from H is lower than the signal amplitude from G (recovering plant) for all frequencies, making it impossible to say which plant is which while judging only from electrophysiological data alone.
In my opinion, the statement on page 3 and in the conclusions (page 14) about differentiating plant health status from the electrophysiological data must be either withdrawn or based on indisputable statistical evidence.
Figures
Fig. 1.
I am not sure what the reasoning is behind highlighting correlations only between some particular sites and not others. If I understand it correctly, the idea was to add more information to a scheme whose main purpose is to show the distances between the sites. However, the superimposed correlation data begs a question: what message do the authors wish to convey by selecting these correlations in particular?
Personally, I find the idea of depicting correlations with arrows that have only one arrowhead a bit misleading, since an arrow signifies unidirectional movement or influence, while a correlation by definition is directionless. I think simple lines or arrows with arrowheads on both ends would be more suitable.
Figs. 7-12.
Y-axis should have not only the parameter stated, but also its units
Specific comments:
Page 2:
The same nomenclature for stating a value and its units should always be used: while it is written “36 ± 2 mΩm−1” (with a space between the value and units), the authors also write “78 ± 2pF mm-1” and “78 ± 400nH m-1” (without a space).
Page 3:
The authors should use the term “flavescence dorée” consistently – should it be in italics (flavescence dorée) or not (flavescence dorée)? Should the first letter be capitalized (Flavescence dorée) or not (flavescence dorée)?
Page 3:
“Scaphoideus titanus” should be in italics
Page 5:
“The first row shows correlations whose absolute value is below 0.3. The only exception is the correlation between B and D, with a negative coefficient of -0.3728 (-0.33021) “. I do not think this is a correct statement because B and D are not on the same row.
Page 6:
In the Fig. 3 legend, it should be “histograms”, not “hystograms”,
Page 10:
“The training was performed by minimizing the validation loss (the loss of the”: the end of the sentence seems to be missing.
Overall, I see a great improvement from the last version. While there are still some concerns that need to be addressed, I recommend the editor to consider publishing this article after a major revision.
P.s. I can clarify that in calling the presented research “unorthodox”, I was complimenting the authors by referring to their research as novel because “orthodox” can have a meaning not only in a religious context: Oxford English Dictionary, Definition 2b:
In extended use: maintaining opinions or practices in accordance with those prevailing or officially sanctioned in one's profession, discipline, party, etc.; conventional, conservative.
Author Response
Point by point response letter
Alessandro Chiolerio*a,b, Federico Tarantoc, Giuseppe Piero Brandinod
aBioinspired Soft Robotics, Istituto Italiano di Tecnologia, Genova, Italy
bUnconventional Computing Laboratory, UWE, Bristol, UK
cHopeful Futures, Zaandam 1507EL, Noord-Holland, The Netherlands
deXact-Lab srl, Via Crispi 56, 34126 Triste, Italy
To the kind attention of the Editor, we wish to express our sincere gratitude
for the time dedicated to our manuscript. In the following, all of the
comments received and our responses are listed.
1. Referee 1
This is a resubmitted manuscript “Towards LocalWeather Forecasts from
Vitis vinifera Electrome” by Alessandro Chiolerio et al. I reviewed the initial
submission and recommended it to be rejected by Biomimetics. In my opinion,
the resubmitted work is vastly improved, the majority of my comments
have been addressed. Despite all the positives, I still have a few observations.
Dear Referee 1, it has been a pleasure to work with you, your suggestions
and doubts moved us to improve the paper, and we wish to acknowledge this.
We believe this new set of changes will clear out the way to publication.
General comments:
Methods
I still fail to find the authors’ interpretation of the meaning behind the
specific correlations distinguished. Is it important that the signals from the
last row of vines (F, G, H) are clustered, but it is not the case for the first row
(A, B, C)? What was the scientific question that the authors solved using
these correlations?
Highlighting the correlations between different plants is fundamental to
understand how redundant is data fed into the ML model. Reducing the
input allows us to invest less time and raw materials. In plant health
monitoring, correlated electrical responses can precede visible symptoms of
stress (drought, pathogens, nutrient deficiency). In dense plant communities
(greenhouses, forests, crops), correlated electrome patterns might indicate
collective behaviors—shared defense responses, synchronized flowering signals,
or ecosystem-level stress adaptation. trong correlations mean more
reliable signal processing, enabling plants themselves to become part of a
cyber-physical sensing infrastructure. Moreover, highlighting the correlations
has become an important part of the exchanges between us and the
referees. A clarification has been added to the text.
The authors added another figure (Fig. 5) to justify the claim that “the
qualitative features of biopotential signals well characterize the health status
of the vines, and we can easily cluster the ”cleaner” waveforms collected
from healthy and plants undergone full recovery, that bring fruits, from dead
stumps still showing an elaborated electrical activity“ (page 3). Looking at
the data (In Fig. 5 and others), I can agree that the control trace (I) and
the diseased plant trace (E) qualitatively differ from other traces. However,
I still see no convincing evidence that it is possible to reliably differentiate
between the plants of different health status, especially between dead
stumps and recovering plants. In their response to my previous comments,
the authors claimed that “The clustering might be influenced by the roots in
anastomosys”, which may be true, but then one cannot be sure which trace
belongs to the dead stump, and which trace belongs to another plant (for
example, vines G and F). The frequency analysis, while commendable, also
shows that, for example, the signal amplitude from the healthy plant (H) is
higher than the signal amplitude from B (recovering plant) for all frequencies.
At the same time, the signal amplitude from H is lower than the signal
amplitude from G (recovering plant) for all frequencies, making it impossible
to say which plant is which while judging only from electrophysiological data
alone. In my opinion, the statement on page 3 and in the conclusions (page
14) about differentiating plant health status from the electrophysiological
data must be either withdrawn or based on indisputable statistical evidence.
We disagree. By combining all of the information available, and carefully
considering the existing correlations between signals, in particular between
vines F and G, it is possible to resolve the health condition of each plant. A
healthy plant features an asymmetric histogram (”H symm” = ”a”) with a
narrow voltage range (”H range” = ”-”), as shown in Figure 3. This feature
is also present in vine B in recovery status with fruits, however all vines in
this status feature also a positive median value (above 0 mV as per Figure
4, ”Med” = ”+”). Stump C features also an asymmetric histogram but
the range of values covered is extended (”H range” = ”+”) and the median
value is negative (”Med” = ”-”). Stump F features are identical to vine
G in recovery status with fruits, with extremely high positive meaningful
correlations (Figure 5) and we can consider this stump as governed in its
electrical response from vine G itself. Other unique features that complete
the sorting are the flat frequency response of the plant in recovery without
fruits, vine A (”FFT” = ”flat”) plus a symmetric histogram (”H symm” =
”s”), the very broad range of values of the histogram of vine E with ongoing
Flavescence dor´ee (”H range” = ”+++”) plus its peculiar frequency response
(”FFT” = ”lin+flat+lin”), and the narrowest voltage range of the control
signal (”H range” = ”—”). Here is a new Table, added to the main text with
the above comments:
__________________________________________________________________________
ID Health status H symm H range Med Avg VS Med FFT
==========================================================================
A recovery s - - = flat
B recovery + fruits a - + > lin+flat
C stump a + - > lin+flat
D recovery + fruits a + + < lin+flat
E flavescent s +++ - = lin+flat+lin
F stump s + + < lin+flat
G recovery + fruits s + + < lin+flat
H healthy a - - < lin+flat
I control a --- - = lin+flat
==========================================================================
Figures Fig. 1. I am not sure what the reasoning is behind highlighting
correlations only between some particular sites and not others. If I understand
it correctly, the idea was to add more information to a scheme whose
main purpose is to show the distances between the sites. However, the superimposed
correlation data begs a question: what message do the authors wish
to convey by selecting these correlations in particular? Personally, I find the
idea of depicting correlations with arrows that have only one arrowhead a
bit misleading, since an arrow signifies unidirectional movement or influence,
while a correlation by definition is directionless. I think simple lines or arrows
with arrowheads on both ends would be more suitable.
Figure 1 was updated with double arrowheads.
Figs. 7-12.
Y-axis should have not only the parameter stated, but also its units
Thank you for noticing it. The units have been added to all the graphs
from Figs 7 to 12.
Specific comments:
Page 2: The same nomenclature for stating a value and its units should
always be used: while it is written “36 ± 2 mΩm1” (with a space between
the value and units), the authors also write “78 ± 2pF mm-1” and “78 ±
400nH m-1” (without a space).
Space added.
Page 3: The authors should use the term “flavescence dor´ee” consistently
– should it be in italics (flavescence dor´ee) or not (flavescence dor´ee)? Should
the first letter be capitalized (Flavescence dor´ee) or not (flavescence dor´ee)?
We opted for italics with first letter capitalized.
Page 3: “Scaphoideus titanus” should be in italics
Thank you for spotting this, now it is reported in italics.
Page 5: “The first row shows correlations whose absolute value is below
0.3. The only exception is the correlation between B and D, with a negative
coefficient of -0.3728 (-0.33021) “. I do not think this is a correct statement
because B and D are not on the same row.
Thank you for spotting this, now amended.
Page 6: In the Fig. 3 legend, it should be “histograms”, not “hystograms”,
Amended.
Page 10: “The training was performed by minimizing the validation loss
(the loss of the”: the end of the sentence seems to be missing.
The sentence is now complete, as follows: ’The training was performed
by minimizing the validation loss (the loss of the validation dataset) and also
the R2 coefficient was collected.’
P.s. I can clarify that in calling the presented research “unorthodox”, I
was complimenting the authors by referring to their research as novel because
“orthodox” can have a meaning not only in a religious context: Oxford
English Dictionary, Definition 2b: In extended use: maintaining opinions or
practices in accordance with those prevailing or officially sanctioned in one’s
profession, discipline, party, etc.; conventional, conservative.
We misunderstood the meaning of the comment provided in the previous
report, thank you for the clarification.

Reviewer 2 Report (New Reviewer)
Comments and Suggestions for Authors
The work is an interesting approach to the subject of plant electrophysiology. The research presented in this paper is very important and could contribute to the development of future precision agriculture methods. However, the paper requires improvement.
The title should be changed to reflect the content of the paper.
Currently, the role of plant electrophysiology is to investigate plant signals in relation to environmental factors. The ability to assess environmental conditions based on plant electrical signals is premature and requires more research.
The Materials and Methods chapter requires a better text structure and separation of individual stages.
Were control trials of signal recording from the apparatus (in the field) without connection to the plants performed? The paper should clearly present the control signal and the signals from the plants.
At the current stage of research on plant signals, specific signal patterns typical of plants should be presented and explained.
The last sentence of abstract is too anthropomorphic for plant activity.
The bibliography is too limited and the introduction does not include the most important findings of electrophysiology and research on the plant electrome.
Author Response
Point by point response letter
Alessandro Chiolerio*a,b, Federico Tarantoc, Giuseppe Piero Brandinod
aBioinspired Soft Robotics, Istituto Italiano di Tecnologia, Genova, Italy
bUnconventional Computing Laboratory, UWE, Bristol, UK
cHopeful Futures, Zaandam 1507EL, Noord-Holland, The Netherlands
deXact-Lab srl, Via Crispi 56, 34126 Triste, Italy
To the kind attention of the Editor, We wish to express our sincere gratitude
for the time dedicated to our manuscript. In the following, all of the
comments received and our responses are listed.
1. Referee 2
The work is an interesting approach to the subject of plant electrophysiology.
The research presented in this paper is very important and could
contribute to the development of future precision agriculture methods. However,
the paper requires improvement. 1. The title should be changed to reflect the content of the paper.
Thank you very much for your comment. We have updated the title to better align
with our study perspectives.
2. Currently, the role of plant electrophysiology is to investigate plant
signals in relation to environmental factors. The ability to assess environmental
conditions based on plant electrical signals is premature and requires
more research.
We agree with the Referee, this is what we believe our preliminary results
in terms of weather forecast might induce other co-workers and colleagues
to explore in the next future. Nevertheless the titles of a couple of papers
we cited are very clear on this regard, for example: ”Electrophysiological
assessment of plant status outside a Faraday cage using supervised machine
learning” and, most significantly, ”Water status assessment in grapevines
using plant electrophysiology”.
3. The Materials and Methods chapter requires a better text structure
and separation of individual stages.
Thank you for spotting this. We have added another layer of structuration
to better separate the contents and divided chapter 2 into 3 sub-chapters,
one of those (2.3) is new and was added in response to the other Referee.).
4. Were control trials of signal recording from the apparatus (in the
field) without connection to the plants performed? The paper should clearly
present the control signal and the signals from the plants.
As per previous interactions with one of the referees, we have collected a
control signal using a cut log of a vine, isolated from the ground to avoid any
form of electrical active or passive connection. This is labelled as individual
”I” in all analyses and graphs, and described in the text. Non connected
cables would have generated spurious signals, acting as floating voltage antennas.
5. At the current stage of research on plant signals, specific signal patterns
typical of plants should be presented and explained.
We believe this was done by several other authors, and as per your last
comment on the introduction, we have incorporated some meaningful references
that discuss such matter. A complete review of signals is outside of the
scopes of the present paper, in particular we refer to a review work published
some months ago that contains all such information. Additional references
are marked in red.
6. The last sentence of abstract is too anthropomorphic for plant activity.
Thank you for spotting this, we agree and have rephrased the sentence,
using a jargon that can be found even in other literature papers.
7. The bibliography is too limited and the introduction does not include
the most important findings of electrophysiology and research on the plant
electrome.
As per previous comment on the discussion of signal patterns, we have
incorporated several works that could better complete our literature analysis.
Additional references are marked in red.

Round 2
Reviewer 1 Report (Previous Reviewer 1)
Comments and Suggestions for Authors
This is a second report on a resubmitted and now amended manuscript “Towards health status determination and local weather forecasts from Vitis vinifera electrome” by Alessandro Chiolerio et al. Much has been improved and a lot of my comments were thoroughly addressed. Unfortunately, I still have a few reservations which concern the claim by the authors that electrophysiological characteristics of the bioelectrical signals from the vines can help determine their health status – a concern that I do not believe was satisfactorily addressed since my last report.
As far as I understand, all vines in principle could be placed in a continuum from being completely healthy to being dead stumps. If the authors aim to employ electrophysiological characteristics that could be useful for determining the health status of a plant (e.g. to have some predictive value), I would expect a presentation of a parameter X (or a set of parameters) that also exists on a continuum and can be directly associated with the health status of a vine in a sense that, for instance, if X value is larger, a plant is healthier. I think it would not be unreasonable to expect that such associations should be theoretically justified or at least discussed.
The authors do provide a key (the table, which, I believe should have a title (Table 1) and a caption) how they link vine health status to electrophysiological signal features, but I find number of problems with these explanations.
“H range” seems to be a parameter the most in line with what I have suggested above: the healthier the plant, the smaller the parameter value (and vice versa). There are still exceptions because recovering plants can be either “+” or “-“ (e.g. D and B).
“H symm” parameter is more problematic. The authors have to state in the methods section how did they evaluate which distributions (histograms) were symmetric and which were not, to make the method indisputable and reproducible. One could argue that left-extending tails of F and G vines make these distributions asymmetric. How do the authors justify their classification? The parameter also seems ambiguous because both the healthy individual (H) and a stump (C) show asymmetric distributions.
Technically, the same argument of reproducibility can be extended to “H range” (what is the cutoff H range for a vine to become classified not as “-“, but as “+”?), though in this case from the visual inspection I am in an agreement with the authors.
I am quite confused by the values in the column “Avg VS Med” because in my understanding when an average and a median of a distribution are roughly equal, these distributions are suggested to be symmetric. However, in the case of, for instance, control (I), the “H symm” states that it is asymmetric, even though median is shown to be equal to the average. Vice versa, F possesses unequal average and median, but it is classified as having a symmetric distribution.
To dispel doubts about the association between the parameter values and plant health status, I think it would be beneficial for the authors to interpret the results: for example, why should a plant in recovery exhibit a voltage distribution with a positive median? I am also not against an idea that not a single parameter but a combination of them reveals the health status, but then again – I see no interpretation why such a particular combination is expected.
Probably the greatest dissatisfactory point to me is, to cite the authors, that “stump F features are identical to vine G in recovery status with fruits”. The authors provide a clear physiological explanation which is satisfactory. However, this very fact that post factum explanations are needed, in my mind completely undermines the approach of unambiguously classifying vines based on their electrophysiological characteristics. What I mean is that if one examines only electrophysiological signals, because of root connectivity one is unable to be sure whether a particular vine is a stump, or recovering. Extending the argument, a possibility cannot be dismissed that any other signals from the vines were influenced by other neighbouring vines, further complicating the interpretation.
Overall, I am very sorry, but I am not convinced. I am not against publishing this article in principle and I see value in what the authors are trying to accomplish. All other inquiries that I have posed are in essence addressed, and while I agree that certain vines display distinct electrophysiological characteristics, the proposed way of identifying vine health status in my opinion is too ambiguous to have predictive value. I think that the report still lacks evidence of unequivocal association between the health status of the vines and their electrophysiological characteristics, thus I believe either more robust evidence must be provided or the claim about this association has to be rescinded. Thus, I advise the editor to propose the article to have a major revision.
Let my comments not overshadow the huge amount of work that has already been accomplished.
Best of wishes to the authors!
Author Response
Response provided in the .pdf letter.

Round 3
Reviewer 1 Report (Previous Reviewer 1)
Comments and Suggestions for Authors
This is a third report on a resubmitted and now amended manuscript “Towards health status determination and local weather forecasts from Vitis vinifera electrome” by Alessandro Chiolerio et al.
I must thank the authors for reminding me that the main topic of the article is centred on the employment of the ML algorithm and not electrophysiology per se, on which my comments have been mainly focused.
In text of the section 2.3, the authors have left the information that was relevant in the previous version but no longer applicable because the table was refined (e.g. mentioning parameter “H symm” or the possible value of H range “---" that is no longer used). Also, the text above the table and below the table gives the same information (e.g. what are the characteristics of the healthy plant), the text below being a more in-depth version. This section should be revised to avoid confusion and duplication.
There are also some remaining reservations from the previous versions. The authors have avoided the issue of F and G vines, which, despite displaying almost identical electrophysiological characteristics, are of very different health status, thus undermining the predictive value of the approach. Thus, a researcher analysing an electrophysiological recording cannot tell whether it comes from a stump or from a recovering plant. Consequently, I do not see the statement “a dead stump is characterized by the unique features…” as true, because, looking from a practical point of view, signals from 2 stumps were registered, and the characteristics of one of them were identical to the characteristics of a recovering plant with fruits. This feature might not affect the later research centred on the ML algorithms, but in my mind it makes it impossible to determine the plant health status from the electrophysiological recordings only.
Philosophically, I am not sure about the reliability of this health-determination key, as it has been constructed using only a very limited number of specimens (e.g. one healthy plant only). I am not convinced that, for example, 100 healthy plants from the same vineyard would all possess unique electrophysiological characteristics that will not overlap with the characteristics of other plants in different states.
The authors seem to be unmoved by my arguments regarding whether the proposed approach really “well characterizes the health status of the vines” and whether the proposed features are really unique. I believe that one reviewer, preoccupied with his reservations, should not gatekeep the science, so I appeal to the editor to decide whether the issues stated above are of importance and should be addressed or not.
And, once again, I must thank the authors for their patience and devotion to improving the presentation of their research!
Author Response
Please see the attachment.

This manuscript is a resubmission of an earlier submission. The following is a list of the peer review reports and author responses from that submission.
Round 1
Reviewer 1 Report
Comments and Suggestions for Authors
The submitted brief report “Towards Local Weather Forecasts from Vitis vinifera Electrome” by Alessandro Chiolerio et al. attempts to apply machine learning models to analyse registered plant electrical activity and then predict local patterns of temperature and humidity fluctuations. This approach is novel and exciting, but there are some concerns that must be addressed before publishing this report.
General comments:
Introduction
I would advise against citing preprints that have not been peer-reviewed yet, meaning the article Wilkening (2024), especially since there is ample literature concerning plant electrical signals. I would rather suggest authors to cite some article from Vladimir Sukhov’s lab, such as this one: https://doi.org/10.3390/plants14101500. Alternatively, Gustavo Maia Souza and his team have written a lot about plant electrome, such as here: https://doi.org/10.1007/s40626-019-00145-x. There are of course other already peer-reviewed options as well.
Personally, I do not think that the last sentence of the introduction (Lines 74-76) adds any value to the article (especially if it is a brief report), because it is expected that any experimental report should have parts detailing methods, results and conclusions.
Methods
While it is a brief report, I believe that the methods must be presented as clearly as possible to erase any possible doubts about the collected results and their interpretation. Unfortunately, I believe many aspects remain to be clarified.
More details about the registration process must be provided: what was the season, were the weather patterns observed typical to the region, or not. The duration of data collection is not explicitly stated in the Methods section. Was it for only 242 hours, as told in Fig. 3? More than 300 hours, as told in Fig. 4? What were the criteria to confirm that the models worked well? How do they compare to the currently used models? What is a threshold for a useful model?
I also have questions about the experimental design: why were precisely these plants selected? How did their health status influence the model? Was the input from dead stumps into the models as important as that from healthy plants? How many plants are needed to develop a practical model?
The description of the plants used is rather vague (lines 90 and onwards).
- How did the authors determine the plant health status?
- “Recovery” after flavescence dorée seems a very generic term: a plant can recover on its own, or some remedies might have been applied. In addition, a plant can have started recovering only very recently, or it can have been infected a long time ago and successfully recovered so no symptoms are visible.
- Does “plant under recovery phase with no fruits yet” mean that the plant is too sick to bear fruits at all, or is the fruiting period simply delayed by the sickness?
- “Healthy individual” is also ambiguous, if the feature of bearing fruits is important.
If a “dead stump” exhibits bioelectrical activity, it is definitely not dead. On the other hand, if a stump is really dead, the electrical activity is definitely not of a biological origin. To dispel these doubts, I would advise the authors to include a control measurement of a definitely dead wooden material – a branch or a log of a vine that has been unequivocally dead, - place it into the ground and insert identical electrodes. This would clearly demonstrate whether the observed electrical activity is of biological origin and whether the “dead” stumps can be called dead.
Such a control would also dispel another misgiving: regarding the registered signals, I am not entirely convinced that the authors are registering meaningful bioelectrical signalling which overpowers the electrical noise of the registration system. For example, the apparent bioelectrical activity can be caused by mechanical vibrations unrelated to biology (see https://doi.org/10.1016/j.funeco.2023.101326). Of course, it is not a problem if the system is still able to accomplish its task, in this case, to predict the weather patterns, but it casts doubt on the biological origin of the signals.
I am not convinced that “the qualitative features of biopotential signals well characterizes the health status of the vines” (line 95 and onwards) because the evidence is not shown. If this statement is supposed to be backed by the data in Fig. 2, it is impossible to verify because each recording is presented in different scales. The dendrogram in Fig. 3 also clusters plants of different health status together (e.g. C and D). If the authors wish to keep this claim, they must present statistical analysis. I imagine looking into the signal/noise ratio or analyzing these signals in the frequency domain might provide some useful information.
Some correlations are distinguished (line 102 and onwards), but the reason for this is not explained. Do the authors suggest that nearby plants have similar electrical activity? Or plants with the same health status? Moreover, B and D are not correlated – they are anticorrelated, so are E and D. p-values are not presented for any of these correlations; thus, one has no idea whether any of them are significant. Also, if the data contain outliers (and it is told so in the Fig. 3 legend and shown in the boxplots), Spearman, rather than Pearson correlation should be calculated.
It is not clear which software was used to analyze the data statistically and which algorithm was used to prepare the dendrogram in Fig. 3.
I cannot comment on the part describing machine learning models because I am not a specialist in that area.
Figures
Fig. 1.
The photo in Fig. 1 is not very helpful, since, if I understand it correctly, because it does not show individuals G, F, and D.
The scheme shown in Fig. 1 is widely inaccurate and should be updated. For example, two squares should equal approximately 1 m, but the distance between E and C is told to be 220 cm (line 108), which is definitely not approximately 1 m.
Fig. 2.
As mentioned above, I find it impossible to compare the traces presented in Fig. 2. The scales of all the traces should be equal. For example, trace A shows a lot more noise than trace G, but then one notices that trace A shows activity whose amplitude (ptp) is ~0,5 mV, while the amplitude (ptp) for trace G is ~6 mv. Thus, the noise levels might or might not be similar - this is extremely unintuitive to judge.
The y-axis of Fig. 2 should have not only the measurement units, but also the parameter that is being measured (e.g. voltage). The time scale is not very intuitive – I recommend changing it to a more reader-friendly way, for example, showing it in minutes or hours, because to understand how much 10^5 seconds really lasts one has to calculate.
Fig. 3.
Fig. 3 is very reader-unfriendly. The letters and numbers are just too small to be readable without zooming in 5 times or so. The panels should be denoted in a more helpful manner, for example A), B), and so on.
Histograms (“histograms”, not “hystograms”), have no label on the x-axes, the bin sizes and number of them appear random, the scales of the y-axes of the histograms are uneven, so it is hard to visually compare the data.
I am not convinced that all the outliers in the boxplot are shown, because for C, the upper part is cut off by the legend, while E, F and G seem to extend beyond -50 mV.
The color coding is inconsistent in the figures: for example, In Fig. 2, the signal from C is orange, and in Fig. 3, it is blue.
It is not stated in the Methods or the Results, what is the reason for preparing the dendrogram, and what information does it tell.
Fig. 4 and Fig. 5.
Fig. 4 and Fig. 5 lack axis labels. “Test_y” in the legend is not helpful at all.
Fig. 6 and Fig. 7.
The axis labels are also missing from Fig. 6 and Fig. 7. The fonts are too small. If the number of epochs was 1200 (Lines 149-150), why in Fig. 6 only 900 are shown? What do the labels “epoch_loss” and “tag:epoch_loss” mean? What does the shaded area behind the signals represent? Why is the start of the recording in Fig. 7 cut off (the blue trace is not visible from the start)?
Results
What I see in Fig. 4 and Fig. 5 are clear 24-hour cycles, which are expected from both bioelectric potentials and temperature and humidity fluctuations. I imagine that this low-frequency component (basically a sine with a 24 h period) is quickly detected by the algorithm, but for other higher-frequency components, I do not see a strong correlation. In other words, I agree that the algorithm can predict that days will be warmer than nights, but more detailed predictions are precise as often as not. In fact, if the authors claim that they “nail” the correlations, they must prove it statistically. In addition, it is still unclear how good the models are because all the values are normalized. I think it would be far more useful to tell the readers how well the models are able to predict temperature and humidity in absolute values: are the errors approximately 2 or 20 °C?
Specific comments:
Lines 19-21:
It is told that machine learning algorithms were “trained to infer short-term meteorological parameters such as temperature, humidity, and atmospheric pressure”. However, the atmospheric pressure is not mentioned in the main text at all.
Lines 22-23:
It is told that “The models demonstrated predictive capabilities with accuracy comparable to sensor-based benchmarks”. However, I did not find any comparison in the main text.
Line 28
The authors state that their method is “non-invasive”, but I do not agree that placing metal electrodes inside a plant (“puncturing the bark”, line 83) can be described as such.
Line 95
Should be characterize (because its plural – “features characterize”)
Line 154
To “nail the correlation” does not sound very scientific.
Lines 161-164
What are the “loss” and “val_loss”?
Line 164
Vitis vinifera should be in italic.
Overall, I found the report to be very poorly prepared, and there is a severe lack of attention to detail. Although it is a brief report, it should still be as rigorously prepared as any type of scientific communication. However, the methods section is severely lacking (at least as far as it concerns recording bioelectrical signals and the selection of the plants): I could not repeat the study based only on the information provided. Moreover, I am not convinced that what the authors have registered is more than just electrical noise because there is no negative control.
A prediction based on plant bioelectrical signals that weather patterns follow a 24-hour cycle does not seem very helpful or practical. I would advise the authors to try to look at day-time and night-time separately, justify their observations statistically, and compare the results to already established models.
In principle, if the goal is to predict weather patterns from plant behavior, in my opinion, a non-invasive technique, such as chlorophyll fluorescence measurement, would be highly preferable.
Overall, while I admire the authors’ courage and dedication to attempt novel and unorthodox research, I do not think that this brief report meets the standards of a Q1 journal, therefore, I recommend the editor to reject it.
Let my critique not discourage them - best of luck for the authors!
Reviewer 2 Report
Comments and Suggestions for Authors
The manuscript by Chiolerio et al. considers using of elecrome for measurement of environmental temperature and humidity changes. The investigation is confused. I have some comments.
(1) My main doubt is related to unclear aim of the work. If authors developed method of prediction of future weather (temperature and humidity), it can be interesting. In this case, it is not clear: What was time interval for prediction of weather on basis of the electrome analysis (one day, two days, etc.)? However, it seems that authors estimated temperature and humidity (in relative units) on basis of analysis of electrome data. If the last variant was shown, importance of this work is not clear. Temperature and humidity sensors are simple technical systems; measurements of these parameters on basis of plant electrical activity are not practically useful (especially in relative units). On the other hand, sensitivity of plants to temperature and humidity seems to be widely-known fact. Maybe, investigation in details of electrical changes in plants under varying temperature and humidity is potentially interesting but this investigation seems to be absent in this work.
(2) Authors investigated both health and infected plants. Why it was investigated? How was infection related to weather prediction? Was investigation of infected plants necessary to weather prediction? It should be described in detail.
(3) Figure 1: Scheme of electrical chain at measurements should be described in detail.
(4) Figure 3: Captions are too small.